# Local Explanation of Dialogue Response Generation

**Yi-Lin Tuan[1], Connor Pryor[2], Wenhu Chen[1], Lise Getoor[2], William Yang Wang[1]**
[1] University of California, Santa Barbara
[2] University of California, Santa Cruz
{ytuan, wenhuchen, william}@cs.ucsb.edu
{cfpryor, getoor}@ucsc.edu

## Abstract

In comparison to the interpretation of classification models, the explanation of sequence generation models is also an important problem, however it has seen little attention. In this work, we study model-agnostic explanations of a representative text generation task – dialogue response generation. Dialog response generation is challenging with its open-ended sentences and multiple acceptable responses. To gain insights into the reasoning process of a generation model, we propose a new method, local explanation of response generation (LERG), that regards the explanations as the mutual interaction of segments in input and output sentences. LERG views the sequence prediction as uncertainty estimation of a human response and then creates explanations by perturbing the input and calculating the certainty change over the human response. We show that LERG adheres to desired properties of explanation for text generation, including unbiased approximation, consistency, and cause identification. Empirically, our results show that our method consistently improves other widely used methods on proposed automatic- and human- evaluation metrics for this new task by 4.4-12.8%. Our analysis demonstrates that LERG can extract both explicit and implicit relations between input and output segments. [1]

## 1 Introduction

As we use machine learning models in daily tasks, such as medical diagnostics [6, 19], speech assistants [31] etc., being able to trust the predictions being made has become increasingly important. To understand the underlying reasoning process of complex machine learning models a sub-field of explainable artificial intelligence (XAI) [2, 17, 36] called local explanations, has seen promising results [35]. Local explanation methods [27, 39] often approximate an underlying black box model by fitting an interpretable proxy, such as a linear model or tree, around the neighborhood of individual predictions. These methods have the advantage of being model-agnostic and locally interpretable.

Traditionally, off-the-shelf local explanation frameworks, such as the Shapley value in game theory [38] and the learning-based Local Interpretable Model-agnostic Explanation (LIME) [35] have been shown to work well on classification tasks with a small number of classes. In particular, there has been work on image classification [35], sentiment analysis [8], and evidence selection for question answering [32]. However, to the best of our knowledge, there has been less work studying explanations over models with sequential output and large class sizes at each time step. An attempt by [1] aims at explaining machine translation by aligning the sentences in source and target languages. Nonetheless, unlike translation, where it is possible to find almost all word alignments of the input and output sentences, many text generation tasks are not alignment-based. We further explore explanations over sequences that contain implicit and indirect relations between the input and output utterances.

---

[1] Our code is available at `https://github.com/Pascalson/LERG`.

35th Conference on Neural Information Processing Systems (NeurIPS 2021).

In this paper, we study explanations over a set of representative conditional text generation models – dialogue response generation models [45, 55]. These models typically aim to produce an engaging and informative [3, 24] response to an input message. The open-ended sentences and multiple acceptable responses in dialogues pose two major challenges: (1) an exponentially large output space and (2) the implicit relations between the input and output texts. For example, the open-ended prompt "How are you today?" could lead to multiple responses depending on the users' emotion, situation, social skills, expressions, etc. A simple answer such as "Good. Thank you for asking." does not have an explicit alignment to words in the input prompt. Even though this alignment does not exist, it is clear that "good" is the key response to "how are you". To find such crucial corresponding parts in a dialogue, we propose to extract explanations that can answer the question: *"Which parts of the response are influenced the most by parts of the prompt?"*

To obtain such explanations, we introduce *LERG*, a novel yet simple method that extracts the sorted importance scores of every input-output segment pair from a dialogue response generation model. We view this sequence prediction as the uncertainty estimation of one human response and find a linear proxy that simulates the certainty caused from one input segment to an output segment. We further derive two optimization variations of LERG. One is learning-based [35] and another is the derived optimal similar to Shapley value [38]. To theoretically verify LERG, we propose that an ideal explanation of text generation should adhere to three properties: unbiased approximation, intra-response consistency, and causal cause identification. To the best of our knowledge, our work is the first to explore explanation over dialog response generation while maintaining all three properties.

To verify if the explanations are both faithful (the explanation is fully dependent on the model being explained) [2] and interpretable (the explanation is understandable by humans) [14], we conduct comprehensive automatic evaluations and user study. We evaluate the *necessity* and *sufficiency* of the extracted explanation to the generation model by evaluating the perplexity change of removing salient input segments (necessity) and evaluating the perplexity of only salient segments remaining (sufficiency). In our user study, we present annotators with only the most salient parts in an input and ask them to select the most appropriate response from a set of candidates. Empirically, our proposed method consistently outperforms baselines on both automatic metrics and human evaluation.

Our key contributions are:

- We propose a novel local explanation method for dialogue response generation (LERG).

- We propose a unified formulation that generalizes local explanation methods towards sequence generation and show that our method adheres to the desired properties for explaining conditional text generation.

- We build a systematic framework to evaluate explanations of response generation including automatic metrics and user study.

## 2   Local Explanation

Local explanation methods aim to explain predictions of an arbitrary model by interpreting the neighborhood of individual predictions [35]. It can be viewed as training a proxy that adds the contributions of input features to a model's predictions [27]. More formally, given an example with input features $x = \{x_i\}_{i=1}^{M}$, the corresponding prediction $y$ with probability $f(x) = P_\theta(Y = y|x)$ (the classifier is parameterized by $\theta$), we denote the contribution from each input feature $x_i$ as $\phi_i \in \mathbb{R}$ and denote the concatenation of all contributions as $\boldsymbol{\phi} = [\phi_1, ..., \phi_M]^T \in \mathbb{R}^M$. Two popular local explanation methods are the learning-based Local Interpretable Model-agnostic Explanations (LIME) [35] and the game theory-based Shapley value [38].

**LIME**   interprets a complex classifier $f$ based on locally approximating a linear classifier around a given prediction $f(x)$. The optimization of the explanation model that LIME uses adheres to:

$$\xi(x) = \arg\min_{\varphi}[L(f, \varphi, \pi_x) + \Omega(\varphi)],\tag{1}$$

where we sample a perturbed input $\tilde{x}$ from $\pi_x(\tilde{x}) = exp(-D(x, \tilde{x})^2/\sigma^2)$ taking $D(x, \tilde{x})$ as a distance function and $\sigma$ as the width. $\Omega$ is the model complexity of the proxy $\varphi$. The objective of $\xi(x)$ is to find the simplest $\varphi$ that can approximate the behavior of $f$ around $x$. When using a linear classifier

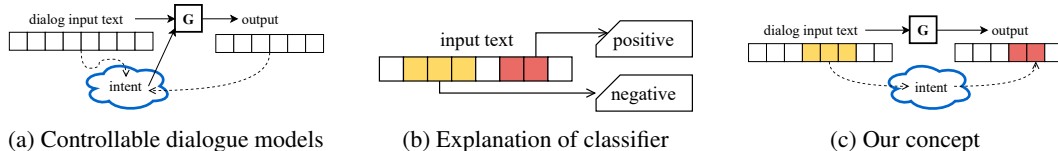

(a) Controllable dialogue models     (b) Explanation of classifier     (c) Our concept

Figure 1: The motivation of local explanation for dialogue response generation. (c) = (a)+(b).

$\boldsymbol{\phi}$ as the $\varphi$ to minimize $\Omega(\varphi)$ [35], we can formulate the objective function as:

$$\boldsymbol{\phi} = \arg\min_{\boldsymbol{\phi}} E_{\tilde{x} \sim \pi_x}(P_\theta(Y = y|\tilde{x}) - \boldsymbol{\phi}^T \mathbf{z})^2, \tag{2}$$

where $\mathbf{z} \in \{0, 1\}^M$ is a simplified feature vector of $\tilde{x}$ by a mapping function $h$ such that $\mathbf{z} = h(x, \tilde{x}) = \{\mathbb{1}(x_i \in \tilde{x})\}_{i=1}^M$. The optimization means to minimize the classification error in the neighborhood of $x$ sampled from $\pi_x$. Therefore, using LIME, we can find an interpretable linear model that approximates any complex classifier's behavior around an example $x$.

**Shapley value** takes the input features $x = \{x_i\}_{i=1}^M$ as $M$ independent players who cooperate to achieve a benefit in a game [38]. The Shapley value computes how much each player $x_i$ contributes to the total received benefit:

$$\varphi_i(x) = \sum_{\tilde{x} \subseteq x \setminus \{x_i\}} \frac{|\tilde{x}|!(|x| - |\tilde{x}| - 1)!}{|x|!}[P_\theta(Y = y|\tilde{x} \cup \{x_i\}) - P_\theta(Y = y|\tilde{x})]. \tag{3}$$

To reduce the computational cost, instead of computing all combinations, we can find surrogates $\phi_i$ proportional to $\varphi_i$ and rewrite the above equation as an expectation over $x$ sampled from $P(\tilde{x})$:

$$\phi_i = \frac{|x|}{|x| - 1}\varphi_i = E_{\tilde{x} \sim P(\tilde{x})}[P_\theta(Y = y|\tilde{x} \cup \{x_i\}) - P_\theta(Y = y|\tilde{x})], \forall i, \tag{4}$$

where $P(\tilde{x}) = \frac{1}{(|x|-1)\binom{|x|-1}{|\tilde{x}|}}$ is the perturb function.[2] We can also transform the above formulation into argmin:

$$\phi_i = \arg\min_{\phi_i} E_{\tilde{x} \sim P(\tilde{x})}([P_\theta(Y = y|\tilde{x} \cup \{x_i\}) - P_\theta(Y = y|\tilde{x})] - \phi_i)^2. \tag{5}$$

## 3 Local Explanation for Dialogue Response Generation

We aim to explain a model's response prediction to a dialogue history one at a time and call it the *local explanation of dialogue response generation*. We focus on the local explanation for a more fine-grained understanding of the model's behavior.

### 3.1 Task Definition

As depicted in Figure 1, we draw inspiration from the notions of controllable dialogue generation models (Figure 1a) and local explanation in sentiment analysis (Figure 1b). The first one uses a concept in predefined classes as the relation between input text and the response; the latter finds the features that correspond to positive or negative sentiment. We propose to find parts within the input and output texts that are related by an underlying intent (Figure 1c).

We first define the notations for dialogue response generation, which aims to predict a response $y = y_1 y_2 ... y_N$ given an input message $x = x_1 x_2 ... x_M$. $x_i$ is the $i$-th token in sentence $x$ with length $M$ and $y_j$ is the $j$-th token in sentence $y$ with length $N$. To solve this task, a typical sequence-to-sequence model $f$ parameterized by $\theta$ produces a sequence of probability masses $<P_\theta(y_1|x), P_\theta(y_2|x, y_1), ..., P_\theta(y_N|x, y_{<N})>$ [45]. The probability of $y$ given $x$ can then be computed as the product of the sequence $P_\theta(y|x) = P_\theta(y_1|x)P_\theta(y_2|x, y_1)...P_\theta(y_N|x, y_{<N})$.

---

[2] $\sum_{\tilde{x} \subseteq x \setminus \{x_i\}} P(\tilde{x}) = \frac{1}{(|x|-1)} \sum_{\tilde{x} \subseteq x \setminus \{x_i\}} 1/\binom{|x|-1}{|\tilde{x}|} = \frac{1}{(|x|-1)} \sum_{|\tilde{x}|} \binom{|x|-1}{|\tilde{x}|}/\binom{|x|-1}{|\tilde{x}|} = \frac{(|x|-1)}{(|x|-1)} = 1.$ This affirms that the $P(\tilde{x})$ is a valid probability mass function.

To explain the prediction, we then define a new explanation model $\Phi \in \mathbb{R}^{M \times N}$ where each column $\Phi_j \in \mathbb{R}^M$ linearly approximates single sequential prediction at the $j$-th time step in text generation. To learn the optimal $\Phi$, we sample perturbed inputs $\tilde{x}$ from a distribution centered on the original inputs $x$ through a probability density function $\tilde{x} = \pi(x)$. Finally, we optimize $\Phi$ by ensuring $u(\Phi_j^T z) \approx g(\tilde{x})$ whenever $z$ is a simplified embedding of $\tilde{x}$ by a mapping function $z = h(x, \tilde{x})$, where we define $g$ as the gain function of the target generative model $f$, $u$ as a transform function of $\Phi$ and $z$ and $L$ as the loss function. Note that $z$ can be a vector or a matrix and $g(\cdot)$, $u(\cdot)$ can return a scalar or a vector depending on the used method. Therefore, we unify the local explanations (LIME and Shapley value) under dialogue response generation as:

**Definition 1: A Unified Formulation of Local Explanation for Dialogue Response Generation**

$$\Phi_j = \arg \min_{\Phi_j} L(g(y_j | \tilde{x}, y_{<j}), u(\Phi_j^T h(\tilde{x}))), \text{ for } j = 1, 2, ..., N . \tag{6}$$

The proofs of unification into Equation 6 can be found in Appendix A. However, direct adaptation of LIME and Shapley value to dialogue response generation fails to consider the complexity of text generation and the diversity of generated examples. We develop disciplines to alleviate these problems.

## 3.2 Proposed Method

Our proposed method is designed to (1) address the exponential output space and diverse responses built within the dialogue response generation task and (2) compare the importance of segments within both input and output text.

First, considering the exponential output space and diverse responses, recent work often generates responses using sampling, such as the dominant beam search with top-k sampling [11]. The generated response is therefore only a sample from the estimated probability mass distribution over the output space. Further, the samples drawn from the distribution will inherently have built-in errors that accumulate along generation steps [34]. To avoid these errors we instead explain the estimated probability of the ground truth human responses. In this way, we are considering that the dialogue response generation model is estimating the certainty to predict the human response by $P_\theta(y|x)$. Meanwhile, given the nature of the collected dialogue dataset, we observe only one response per sentence, and thus the mapping is deterministic. We denote the data distribution by $P$ and the probability of observing a response $y$ given input $x$ in the dataset by $P(y|x)$. Since the mapping of $x$ and $y$ is deterministic in the dataset, we assume $P(y|x) = 1$.

Second, if we directly apply prior explanation methods of classifiers on sequential generative models, it turns into a One-vs-Rest classification situation for every generation step. This can cause an unfair comparison among generation steps. For example, the impact from a perturbed input on $y_j$ could end up being the largest just because the absolute certainty $P_\theta(y_j | x, y_{<j})$ was large. However, the impact from a perturbed input on each part in the output should be *how much the certainty has changed after perturbation* and *how much the change is compared to other parts*.

Therefore we propose to find explanation in an input-response pair $(x, y)$ by comparing the interactions between segments in $(x, y)$. To identify the most salient interaction pair $(x_i, y_j)$ (the $i$-th segment in $x$ and the $j$-th segment in $y$), we anticipate that a perturbation $\tilde{x}$ impacts the $j$-th part most in $y$ if it causes

$$D(P_\theta(y_j | \tilde{x}, y_{<j}) || P_\theta(y_j | x, y_{<j})) > D(P_\theta(y_{j'} | \tilde{x}, y_{<j'}) || P_\theta(y_{j'} | x, y_{<j'})), \forall j' \neq j , \tag{7}$$

where $D$ represents a distance function measuring the difference between two probability masses. After finding the different part $x_i$ in $x$ and $\tilde{x}$, we then define an existing salient interaction in $(x, y)$ is $(x_i, y_j)$.

In this work, we replace the distance function $D$ in Equation 7 with Kullback–Leibler divergence ($D_{KL}$) [20]. However, since we reduce the complexity by considering $P_\theta(y|x)$ as the certainty estimation of $y$, we are limited to obtaining only one point in the distribution. We transfer the equation by modeling the estimated joint probability by $\theta$ of $x$ and $y$. We reconsider the joint distributions as $P_\theta(\tilde{x}, y_{\leq j})$ such that $\sum_{\tilde{x}, y} P_\theta(\tilde{x}, y_{\leq j}) = 1$ and $q(\tilde{x}, y) = P_{\theta, \pi_{inv}}(\tilde{x}, y_{\leq j}) = P_\theta(x, y)$ such that $\sum_{\tilde{x}, y} q(\tilde{x}, y) = \sum_{\tilde{x}, y} P_\theta(x, y_{\leq j}) = \sum_{\tilde{x}, y} P_{\theta, \pi_{inv}}(\tilde{x}, y_{\leq j}) = 1$ with $\pi_{inv}$ being the inverse function

of $\pi$. Therefore,

$$D(P_\theta(\tilde{x}, y_{\leq j})||P_\theta(x, y_{\leq j})) = D_{KL}(P_\theta(\tilde{x}, y_{\leq j})||q(\tilde{x}, y_{\leq j})) = \sum_{y_j} \sum_{\tilde{x}} P_\theta(\tilde{x}, y_{\leq j}) \log \frac{P_\theta(\tilde{x}, y_{\leq j})}{P_\theta(x, y_{\leq j})} \, . \tag{8}$$

Moreover, since we are estimating the certainty of a response $y$ drawn from data distribution, we know that the random variables $\tilde{x}$ is independently drawn from the perturbation model $\pi$. Their independent conditional probabilities are $P(y|x) = 1$ and $\pi(\tilde{x}|x)$. We approximate the multiplier $P_\theta(\tilde{x}, y_{\leq j}) \approx P(\tilde{x}, y_{\leq j}|x) = P(\tilde{x}|x)P(y|x) = \pi(\tilde{x}|x)$. The divergence can be simplified to

$$D(P_\theta(\tilde{x}, y_{\leq j})||P_\theta(x, y_{\leq j})) \approx \sum_{y_j} \sum_{\tilde{x}} \pi(\tilde{x}|x) \log \frac{P_\theta(\tilde{x}, y_{\leq j})}{P_\theta(x, y_{\leq j})} = E_{\tilde{x} \sim \pi(\cdot|x)} \log \frac{P_\theta(\tilde{x}, y_{\leq j})}{P_\theta(x, y_{\leq j})} \, . \tag{9}$$

To meet the inequality for all $j$ and $j' \neq j$, we estimate each value $\Phi_j^T \mathbf{z}$ in the explanation model $\Phi$ being proportional to the divergence term, where $\mathbf{z} = h(x, \tilde{x}) = \{\mathbb{1}(x_i \in \tilde{x})\}_{i=1}^M$. It turns out to be re-estimating the distinct of the chosen segment $y_j$ by normalizing over its original predicted probability.

$$\Phi_j^T \mathbf{z} \propto E_{\tilde{x} \subseteq x \setminus \{x_i\}} D(P_\theta(\tilde{x}, y_{\leq j})||P_\theta(x, y_{\leq j})) \approx E_{\tilde{x}, \tilde{x} \subseteq x \setminus \{x_i\}} \log \frac{P_\theta(\tilde{x}, y_{\leq j})}{P_\theta(x, y_{\leq j})} \, . \tag{10}$$

We propose two variations to optimize $\Phi$ following the unified formulation defined in Equation 6.

First, since logarithm is strictly increasing, so to get the same order of $\Phi_{ij}$, we can drop off the logarithmic term in Equation 10. After reducing the non-linear factor, we use mean square error as the loss function. With the gain function $g = \frac{P_\theta(\tilde{x}, y_{\leq j})}{P_\theta(x, y_{\leq j})}$, the optimization equation becomes

$$\Phi_j = \arg\min_{\Phi_j} E_{P(\tilde{x})} \left( \frac{P_\theta(\tilde{x}, y_{\leq j})}{P_\theta(x, y_{\leq j})} - \Phi_j^T \mathbf{z} \right)^2, \forall j \, . \tag{11}$$

We call this variation as LERG_L in Algorithm 1, since this optimization is similar to LIME but differs by the gain function being a ratio.

To derive the second variation, we suppose an optimized $\Phi$ exists and is denoted by $\Phi^*$, we can write that for every $\tilde{x}$ and its correspondent $\mathbf{z} = h(x, \tilde{x})$,

$$\Phi_j^* \mathbf{z} = \log \frac{P_\theta(\tilde{x}, y_{\leq j})}{P_\theta(x, y_{\leq j})} \, . \tag{12}$$

We can then find the formal representation of $\Phi_{ij}^*$ by

$$\begin{aligned}
\Phi_{ij}^* &= \Phi_j^* \mathbf{1} - \Phi_j^* \mathbf{1}_{i=0} \\
&= \Phi_j^* (\mathbf{z} + e_i) - \Phi_j^* \mathbf{z}, \forall \tilde{x} \in x \setminus \{x_i\} \text{ and } \mathbf{z} = h(x, \tilde{x}) \\
&= E_{\tilde{x} \in x \setminus \{x_i\}}[\Phi_j^*(\mathbf{z} + e_i) - \Phi_j^* \mathbf{z}] \\
&= E_{\tilde{x} \in x \setminus \{x_i\}}[\log P_\theta(y_j|\tilde{x} \cup \{x_i\}, y_{<j}) - \log P_\theta(y_j|\tilde{x}, y_{<j})]
\end{aligned} \tag{13}$$

We call this variation as LERG_S in Algorithm 1, since this optimization is similar to Shapley value but differs by the gain function being the difference of logarithm. To further reduce computations, we use Monte Carlo sampling with $m$ examples as a sampling version of Shapley value [41].

## 3.3 Properties

We propose that an explanation of dialogue response generation should adhere to three properties to prove itself faithful to the generative model and understandable to humans.

**Property 1: unbiased approximation** *To ensure the explanation model $\Phi$ explains the benefits of picking the sentence $y$, the summation of all elements in $\Phi$ should approximate the difference between the certainty of $y$ given $x$ and without $x$ (the language modeling of $y$).*

$$\sum_j \sum_i \Phi_{ij} \approx \log P(y|x) - \log P(y) \, . \tag{14}$$

**Algorithm 1:** LOCAL EXPLANATION OF RESPONSE GENERATION

---

**Input:** input message $x = x_1 x_2 ... x_M$, ground-truth response $y = y_1 y_2 ... y_N$
**Input:** a response generation model $\theta$ to be explained
**Input:** a local explanation model parameterized by $\Phi$
// 1st variation – LERG_L
**for** *each iteration* **do**

    sample a batch of $\tilde{x}$ perturbed from $\pi(x)$
    map $\tilde{x}$ to $z = \{0, 1\}_1^M$
    compute gold probability $P_\theta(y_j | x, y_{<j})$
    compute perturbed probability $P_\theta(y_j | \tilde{x}, y_{<j})$
    optimize $\Phi$ to minimize loss function
        $L = \sum_j \sum_{\tilde{x}} (\frac{P_\theta(y_j | \tilde{x}, y_{<j})}{P_\theta(y_j | x, y_{<j})} - \Phi_j^T \mathbf{z})^2$

// 2nd variation - LERG_S
**for** *each* $i$ **do**

    sample a batch of $\tilde{x}$ perturbed from $\pi(x \backslash \{x_i\})$
    $\Phi_{ij} = \frac{1}{m} \sum_{\tilde{x}} \log P_\theta(y_j | \tilde{x} \cup \{x_i\}, y_{<j}) - \log P_\theta(y_j | \tilde{x}, y_{<j})$, for $\forall j$
return $\Phi_{ij}$, for $\forall i, j$

---

**Property 2: consistency**  *To ensure the explanation model $\Phi$ consistently explains different genera-tion steps $j$, given a distance function if*

$$D(P_\theta(y_j | \tilde{x}, y_{<j}), P_\theta(y_j | \tilde{x} \cup \{x_i\}, y_{<j})) > D(P_\theta(y_{j'} | \tilde{x}, y_{<j'}), P_\theta(y_{j'} | \tilde{x} \cup \{x_i\}, y_{<j'})), \forall j', \forall \tilde{x} \in x \backslash \{x_i\}, \tag{15}$$

then $\Phi_{ij} > \Phi_{ij'}$.

**Property 3: cause identification**  *To ensure that the explanation model sorts different input features by their importance to the results, if*

$$g(y_j | \tilde{x} \cup \{x_i\}) > g(y_j | \tilde{x} \cup \{x_i'\}), \forall \tilde{x} \in x \backslash \{x_i, x_i'\}, \tag{16}$$

then $\Phi_{ij} > \Phi_{i'j}$

We prove that our proposed method adheres to all three Properties in Appendix B. Meanwhile Shapley value follows Properties 2 and 3, while LIME follows Property 3 when an optimized solution exists. These properties also demonstrate that our method approximates the text generation process while sorting out the important segments in both the input and output texts. This could be the reason to serve as explanations to any sequential generative model.

## 4 Experiments

Explanation is notoriously hard to evaluate even for digits and sentiment classification which are generally more intuitive than *explaining response generation*. For digit classification (MNIST), explanations often mark the key curves in figures that can identify digit numbers. For sentiment analysis, explanations often mark the positive and negative words in text. Unlike them, we focus on identifying the key parts in both input messages and their responses. Our move requires an explanation include the interactions of the input and output features.

To evaluate the defined explanation, we quantify the necessity and sufficiency of explanations towards a model's uncertainty of a response. We evaluate these aspects by answering the following questions.

- **necessity:** How is the model influenced after removing explanations?

- **sufficiency:** How does the model perform when only the explanations are given?

Furthermore, we conduct a user study to judge human understandings of the explanations to gauge how trustworthy the dialog agents are.

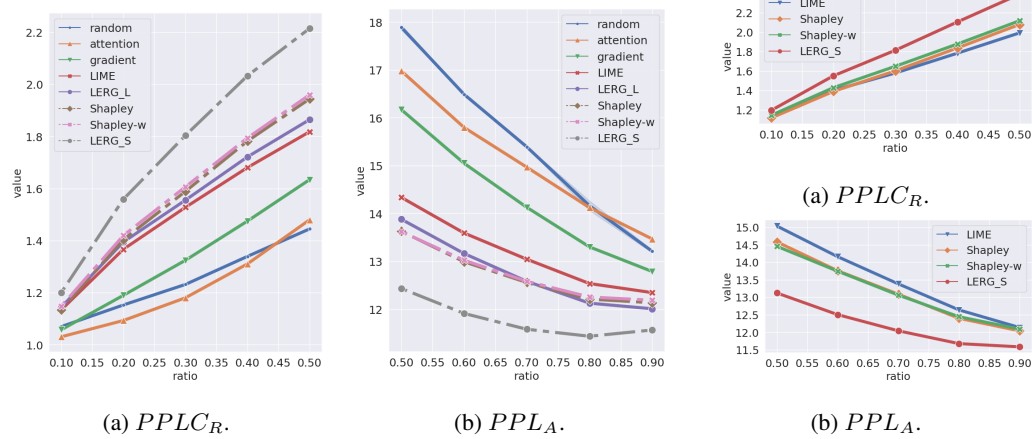

(a) $PPLC_R$.

(b) $PPL_A$.

(a) $PPLC_R$.

(b) $PPL_A$.

Figure 2: The explanation results of a GPT model fine-tuned on DailyDialog.

Figure 3: The explanation results of fine-tuned DialoGPT.

## 4.1 Dataset, Models, Methods

We evaluate our method over chit-chat dialogues for their more complex and realistic conversations. We specifically select and study a popular conversational dataset called DailyDialog [25] because its dialogues are based on daily topics and have less uninformative responses.Due to the large variation of topics, open-ended nature of conversations and informative responses within this dataset, explaining dialogue response generation models trained on DailyDialog is challenging but accessible.[3]

We fine-tune a GPT-based language model [33, 47] and a DialoGPT [55] on DailyDialog by minimizing the following loss function:

$$L = -\sum_m \sum_j \log P_\theta(y_j|x, y_{<j}),\tag{17}$$

where $\theta$ is the model's parameter. We train until the loss converges on both models and achieve fairly low test perplexities compared to [25]: $12.35$ and $11.83$ respectively. The low perplexities demonstrate that the models are more likely to be rationale and therefore, evaluating explanations over these models will be more meaningful and interpretable.

We compare our explanations LERG_L and LERG_S with attention [46], gradient [43], LIME [35] and Shapley value [42]. We use sample mean for Shapley value to avoid massive computations (Shapley for short), and drop the weights in Shapley value (Shapley-w for short) due to the intuition that not all permutations should exist in natural language [12, 21]. Our comparison is fair since all methods requiring permutation samples utilize the same amount of samples.[4]

## 4.2 Necessity: How is the model influenced after removing explanations?

Assessing the correctness of estimated important feature relevance requires labeled features for each model and example pair, which is rarely accessible. Inspired by [2, 4] who removes the estimated salient features and observe how the performance changes, we introduce the notion *necessity* that extends their idea. We quantify the necessity of the estimated salient input features to the uncertainty estimation of response generation by *perplexity change of removal* ($PPLC_R$), defined as:

$$PPLC_R := exp^{\frac{1}{m}[-\sum_j \log P_\theta(y_j|x_R, y_{<j}) + \sum_j \log P_\theta(y_j|x, y_{<j})]},\tag{18}$$

where $x_R$ is the remaining sequence after removing top-k% salient input features.

---

[3]We include our experiments on personalized dialogues and abstractive summarization in Appendix E

[4]More experiment details are in Appendix C

As shown in Figure 2a and Figure 3a[5], removing larger number of input features consistently causes the monotonically increasing $PPLC_R$. Therefore, to reduce the factor that the $PPLC_R$ is caused by, the removal ratio, we compare all methods with an additional baseline that *randomly* removes features. LERG_S and LERG_L both outperform their counterparts Shapley-w and LIME by 12.8% and 2.2% respectively. We further observe that Shapley-w outperforms the LERG_L. We hypothesize that this is because LERG_L and LIME do not reach an optimal state.

## 4.3 Sufficiency: How does the model perform when only the explanations are given?

Even though necessity can test whether the selected features are crucial to the model's prediction, it lacks to validate how possible the explanation itself can determine a response. A complete explanation is able to recover model's prediction without the original input. We name this notion as *sufficiency* testing and formalize the idea as:

$$PPL_A := exp^{-\frac{1}{m}\sum_j \log P_\theta(y_j|x_A, y_{<j})} , \tag{19}$$

where $x_A$ is the sequential concatenation of the top-k% salient input features.

As shown in Figure 2b and Figure 3b, removing larger number of input features gets the $PPL_A$ closer to the perplexity of using all input features $12.35$ and $11.83$. We again adopt a random baseline to compare. LERG_S and LERG_L again outperform their counterparts Shapley-w and LIME by 5.1% and 3.4% respectively. Furthermore, we found that LERG_S is able to go lower than the original $12.35$ and $11.83$ perplexities. This result indicates that LERG_S is able to identify the most relevant features while avoiding features that cause more uncertainty during prediction.

## 4.4 User Study

To ensure the explanation is easy-to-understand by non machine learning experts and gives users insights into the model, we resort to user study to answer the question: "If an explanation can be understood by users to respond?"

We ask human judges to compare explanation methods. Instead of asking judges to annotate their explanation for each dialogue, to increase their agreements we present only the explanations (Top 20% features) and ask them to choose from four response candidates, where one is the ground-truth, two are randomly sampled from other dialogues, and the last one is randomly sampled from other turns in the same dialogue. Therefore the questionnaire requires human to interpret the explanations but not guess a response that has word overlap with the explanation. The higher accuracy indicates the higher quality of explanations. To conduct more valid human evaluation, we randomly sample 200 conversations with sufficiently long input prompt (length$\geq 10$). This way it filters out possibly non-explainable dialogues that can cause ambiguities to annotators and make human evaluation less reliable.

| Method | Acc | Conf |
|---|---|---|
| Random | 36.15 | 3.00 |
| Attention | 34.75 | 2.81 |
| Gradient | 42.52 | 2.97 |
| LIME | 46.37 | 3.26 |
| LERG_L | 47.97 | 3.24 |
| Shapley-w | 53.65 | 3.20 |
| LERG_S | **56.03** | **3.35** |

Table 1: Confidence (1-5) with 1 denotes *not confident* and 5 denotes *highly confident*.

We employ three workers on Amazon Mechanical Turk [7] [6] for each method of each conversation, resulting in total 600 annotations. Besides the multiple choice questions, we also ask judges to claim their confidences of their choices. The details can be seen in Appendix D. The results are listed in Table 1. We observe that LERG_L performs slightly better than LIME in accuracy while maintaining similar annotator's confidence. LERG_S significantly outperforms Shapley-w in both accuracy and annotators' confidence. Moreover, these results indicates that when presenting users with only 20% of the tokens they are able to achieve 56% accuracy while a random selection is around 25%.

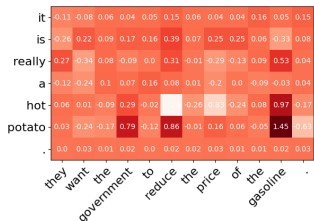
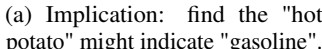
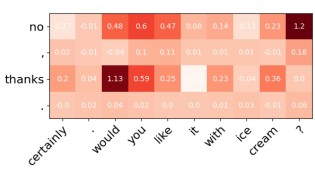
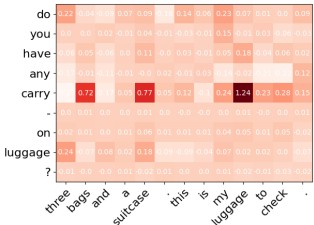

(a) Implication: find the "hot potato" might indicate "gasoline".

(b) Sociability: find "No" for the "question mark" and "thanks" for the "would like", the polite way to say "want".

(c) Error analysis: related but not the best

Figure 4: Two major categories of local explanation except word alignment and one typical error. The horizontal text is the input prompt and the vertical text is the response.

## 4.5 Qualitative Analysis

We further analyzed the extracted explanation for each dialogue. We found that these fine-grained level explanations can be split into three major categories: implication / meaning, sociability, and one-to-one word mapping. As shown in Figure 4a, the "hot potato" in response implies the phenomenon of "reduce the price of gasoline". On the other hand, Figure 4b demonstrates that a response with sociability can sense the politeness and responds with "thanks". We ignore word-to-word mapping here since it is intuitive and can already be successfully detected by attention models. Figure 4c shows a typical error that our explanation methods can produce. As depicted, the word "carry" is related to "bags","suitcases", and "luggage". Nonetheless a complete explanation should cluster "carry-on luggages". The error of explanations can result from (1) the target model or (2) the explanation method. When taking the first view, in future work, we might use explanations as an evaluation method for dialogue generation models where the correct evaluation metrics are still in debates. When taking the second view, we need to understand that these methods are *trying* to explain the model and are not absolutely correct. Hence, we should carefully analyze the explanations and use them as reference and should not fully rely on them.

## 5 Related Work and Discussion

Explaining dialogue generation models is of high interest to understand if a generated response is reasonably produced rather than being a random guess. For example, among works about controllable dialogue generation [15, 26, 37, 40, 48, 50, 51, 53], Xu et al. [49] takes the dialog act in a controllable response generation model as the explanation. On the other hand, some propose to make dialogue response generation models more interpretable through walking on knowledge graphs [18, 28, 44]. Nonetheless, these works still rely on models with complex architecture and thus are not fully interpretable. We observe the lack of a model-agnostic method to analyze the explainability of dialogue response generation models, thus proposing LERG.

Recently, there are applications and advances of local explanation methods [27, 35, 38]. For instance in NLP, some analyze the contributions of segments in documents to positive and negative senti-ments [4, 8, 9, 29]. Some move forwards to finding segments towards text similarity [10], retrieving a text span towards question-answering [32], and making local explanation as alignment model in machine translation [1]. These tasks could be less complex than explaining general text generation models, such as dialogue generation models, since the the output space is either limited to few classes or able to find one-to-one mapping with the input text. Hence, we need to define how local explanations on text generation should work. However, we would like to note that LERG serves as a general formulation for explaining text generation models with flexible setups. Therefore, the distinct of prior work can also be used to extend LERG, such as making the explanations hierarchical. To move forward with the development of explanation methods, LERG can also be extended to dealing

---

[5]We did a z-test and a t-test [22] with the null hypothesis between LERG_L and LIME (and LERG_S and Shapley). For both settings the p-value was less than 0.001, indicating that the proposed methods significantly outperform the baselines.

[6]https://www.mturk.com

with off- /on- data manifold problem of Shapley value introduced in [13], integrating causal structures to separate direct / in-direct relations [12, 16], and fusing concept- / feature- level explanations [5].

# 6    Conclusion

Beyond the recent advances on interpreting classification models, we explore the possibility to understand sequence generation models in depth. We focus on dialogue response generation and find that its challenges lead to complex and less transparent models. We propose local explanation of response generation (LERG), which aims at explaining dialogue response generation models through the mutual interactions between input and output features. LERG views the dialogue generation models as a certainty estimation of a human response so that it avoids dealing with the diverse output space. To facilitate future research, we further propose a unification and three properties of explanations for text generation. The experiments demonstrate that LERG can find explanations that can both recover a model's prediction and be interpreted by humans. Next steps can be taking models' explainability as evaluation metrics, integrating concept-level explanations, and proposing new methods for text generation models while still adhering to the properties.

# 7    Acknowledgement

We thank all the reviewers precious comments in revising this paper. This material is based on work that is partially funded by an unrestricted gift from Google.

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
