# A  Unifying LIME and Shapley value to Dialogue Response Generation

## A.1  Learning based Local Approximation Explanation

Taking the prediction of each time step in sequence generation as classification over a lexicon, we can define the loss function as mean square error, the $u(x) = x$, $\mathbf{z} = h(\tilde{x}) = \{\mathbb{1}(\tilde{x}_i = x_i)\}_{i=1}^{|x|}$ and the gain function $g$ as

$$g(y_j|\tilde{x}, y_{<j}) = P_\theta(y_j|\tilde{x}, y_{<j}) \tag{20}$$

Then LIME can be cast exactly as Equation 6.

$$\Phi_j = \arg\min_{\Phi_j} E_{P(\tilde{x})}(P_\theta(y_j|\tilde{x}, y_{<j}) - \Phi_j^T z)^2, \forall j \tag{21}$$

## A.2  Game Theory based Attribution Methods

We first define the gain function is

$$g(y_j|\tilde{x}, y_{<j})_i = P_\theta(y_j|\tilde{x} \cup i, y_{<j}) - P_\theta(y_j|\tilde{x}, y_{<j}), \text{if } i \notin \tilde{x} \text{ else } g(y_j|\tilde{x}, y_{<j})_i = 0 \tag{22}$$

The $Z = h(\tilde{x}) \in \mathbb{R}^{M \times M}$ is defined to be a diagonal matrix with the diagonal being

$$Z_{ii} = 1, \text{if } i \notin \tilde{x} \text{ else } 0 = 0 \tag{23}$$

With the loss function being L2-norm, we can see the equation is exactly the same as the Equation 6.

$$\Phi_j = \arg\min_{\Phi_j} E_{P(\tilde{x})}||g(y_j|\tilde{x}, y_{<j}) - \Phi_j^T Z||_2 \tag{24}$$

# B  The proof of Properties

**Property 1: unbiased approximation**  *To ensure the explanation model $\Phi$ explains the benefits of picking the sentence $y$, the summation of all elements in $\Phi$ should approximate the difference between the certainty of $y$ given $x$ and without $x$ (the language modeling of $y$).*

$$\sum_j \sum_i \Phi_{ij} \approx \log P(y|x) - \log P(y) \tag{25}$$

*proof:*

$$\begin{aligned}
\sum_j \sum_i \Phi_{ij} &= \sum_j \sum_i E_{\tilde{x} \in S \setminus i}[\log P(y_j|\tilde{x} \cup \{x_i\}) - \log P(y_j|\tilde{x})] \\
&= \sum_j [\sum_i E \log P(y_j|\tilde{x} \cup \{x_i\}) - \sum_i E \log P(y_j|\tilde{x})] \\
&\approx \sum_j [\sum_i \sum_{\tilde{x} \cup \{x_i\}} P(\tilde{x}) \log P(y_j|\tilde{x} \cup \{x_i\}) - \sum_i \sum_{\tilde{x}} P(\tilde{x}) \log P(y_j|\tilde{x})] \\
&= \sum_j [\log P(y_j|x, y_{<j}) - \log P(y_j|\emptyset, y_{<j})] \\
&= \log P(y|x) - \log P(y)
\end{aligned} \tag{26}$$

**Property 2: consistency**  *To ensure the explanation model $\Phi$ consistently explains different generation steps $j$, given a distance function if*

$$D(P_\theta(y_j|\tilde{x}, y_{<j}), P_\theta(y_j|\tilde{x} \cup \{x_i\}, y_{<j})) > D(P_\theta(y_{j'}|\tilde{x}, y_{<j'}), P_\theta(y_{j'}|\tilde{x} \cup \{x_i\}, y_{<j'})), \forall j', \forall \tilde{x} \in x \setminus \{x_i\} \tag{27}$$

then $\Phi_{ij} > \Phi_{ij'}$.

When taking the distance function as the difference between log-probabilities, we can prove that Equation 13 has this property by reducing the property to be the *consistency* property of Shapley value [27]. Prior work [52] has shown that the assumed prior distribution of Shapley value is the only one that satisfies this monotonicity.

Choose a better response to the given masked (not complete) input message. And select a score (1-5) indicates your confidence in this choice.

**For example**, we have an input message: "What would the roses cost me?"

However, the input message is somehow masked and become "__ **roses cost** __"

Could you select the correct response given only the partial input message?

A: They are only $20 a dozen.
B: Of course, I do.
C: Do you have an air sickness?
D: Perhaps you'd be interested in red roses.
In this example, the correct answer is A.

(a) instruction

**1. Masked Input Message:**

wait __ what __ of __

Q1: Choose a better response.
○ I think it s important to do now . We can have a birthday party for you when you come out of the hospital .
○ OK . I will wait for you inside the restaurant
○ Sound great . Let's go !
○ What's the problem ?

Q2: On a scale of 1-5, how much are you confident in your choice? (1: low confident; 5: high confident)

(b) question

Figure 5: The screenshots of our user study

**Property 3: cause identification**   *To ensure that the explanation model sorts different input features by their importance to the results, if*

$$g(y_j|\tilde{x} \cup \{x_i\}) > g(y_j|\tilde{x} \cup \{x_i'\}), \forall \tilde{x} \in x\backslash\{x_i, x_i'\} \qquad (28)$$

then $\Phi_{ij} > \Phi_{i'j}$

The unified formulation (Equation 6) is to minimize the difference between $\phi$ and the gain function $g$. If an optimized $\phi^*$ exists, $g$ can be written as $g(y_j|\tilde{x} \cap i) = \phi_j^*(\tilde{z} + e_{i=1})$. Therefore the inequality 28 can be derived as:

$$\begin{aligned}
&g(y_j|\tilde{x} \cup \{x_i\}) > g(y_j|\tilde{x} \cup \{x_i'\}), \forall \tilde{x} \in x\backslash\{x_i, x_i'\} \\
&\iff \phi_j^*(\tilde{z} + e_i) > \phi_j^*(\tilde{z} + e_{i'}) \\
&\iff \phi_j^* e_i > \phi_j^* e_{i'} \\
&\iff \phi_{ij}^* > \phi_{i'j}^*
\end{aligned} \qquad (29)$$

Since Shapley_log of LERG is a variation to express the optimized $\phi$, the method adheres to Property 3 without assumptions.

## C   Experiment Details

For all methods, we sample 1000 perturbations of input. Also, to reduce the effect of off-manifold perturbations, we perturb input with at most half the features are changed. After three runs with random initialization for each method, the variances of the results are at most 0.001 and do not affect the trend. Our code is available at `https://github.com/Pascalson/LERG`.

All methods can be run on single GPU. We run them on one TITAN RTX.

## D   User Study Details

Figure 5 are the screenshots of our instruction and question presented to workers on Amazon Mechanical Turk. We paid workers an estimated hourly wage 18 usd. For every 5 questions, the estimated spent time is about 30 seconds and the worker was paid 0.15 usd.

## E   More Experiments

Beyond our main experiments on Dailydialog [25], we further take a look into how Shapley value and LERG_S work on personalized conversation and abstractive summarization. We specifically use datasets PersonChat [54] for the personalized conversation task and XSum [30] for abstractive summarization. We fine-tuned a GPT model on PersonaChat as [47] and directly used the BART model [23] that has been pretrained on XSum. The

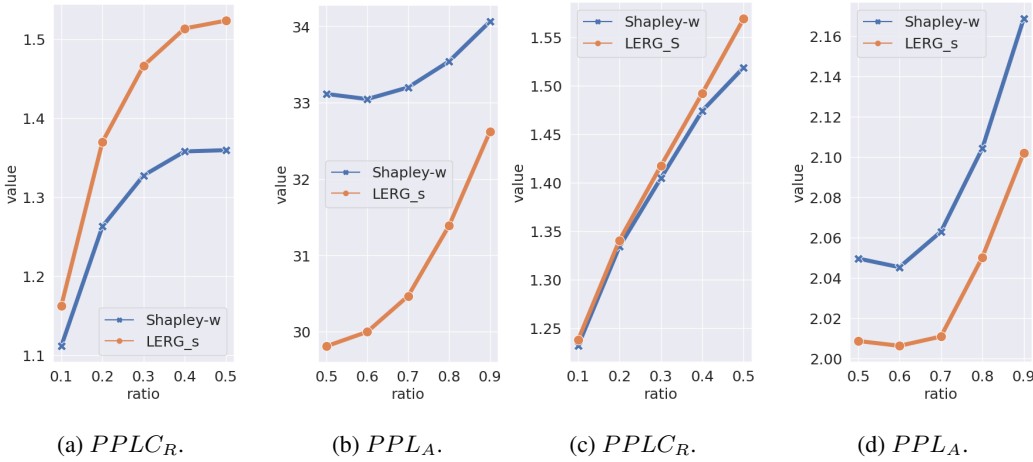

(a) $PPLC_R$.  (b) $PPL_A$.  (c) $PPLC_R$.  (d) $PPL_A$.

Figure 6: The explanation results of a GPT model fine-tuned on PersonaChat ((a) and (b)) and the pretrained BART model on XSum ((c) and (d))

results are plotted in Figure 6. Similar to the results of our main experiments, Figure 6a and 6c show similar trend of increased $PPLC_R$ with a higher removal ratio, with LERG_S consistently performing better. Further, the $PPL_A$ also shows a similar trend to the main experiments, with the perplexity decreasing until some ratio and subsequently increase. Interestingly, the lowest ratio occurs earlier compared to the experiments on DailyDialog. This phenomenon can mean that the number of key terms in the input text of PersonaChat and XSum are less than the one of DailyDialog. Besides this ratio, we observe that LERG_S consistently has lower perplexity.

**Implementation Details**  Throughout the preliminary study of PersonaChat, we specifically investigate the influence of dialogue history to the response and ignore profiles in the input. For XSum, we only run explanation methods on documents containing less than 256 tokens and responses with greater than 30 tokens, therefore, the perplexity is lower than the one reported in [23].

# F  Discussion of Local Explanation with Phrase-level Input Features

We tried two methods of phrase-level experiments. In the first approach, we used parsed phrases, instead of tokens, as the $x_i$ and $y_j$ in the equations and obtained explanations through Algorithm 1. In the second approach, we used the token-level explanations and averaged the scores of tokens in the same phrase parsed by an off-the-shelf parser. In both cases, the trend of the performance was similar to token-level methods. However, we suppose that the basic unit of dialogues is hard to define. Therefore we choose tokens in this paper for that tokens can be flexibly bottom-up to different levels of units.