# OpenReview forum: "Local Explanation of Dialogue Response Generation"
_NeurIPS.cc/2021/Conference — NeurIPS 2021 Poster_

### Official Review · Reviewer_CScg · 2021-07-01

**Rating:** 6
**Confidence:** 4

**Summary:**

The paper addresses the local explanation of sequence generation models, specifically under the dialogue response generation setting, and proposes local explanation of response generation (LERG) that computes the mutual interactions between input and output sequences under perturbations as the explanations. Automatic and human evaluations show that LERG is more effective than other baselines.

Contributions:
1. A new local explanation method for dialogue response generation
2. Such an explanation method could be generalized to other generation models as well
3.  A systematic framework to evaluate the dialogue model explanation quality involving both automatic evaluations and human studies.

**Limitations And Societal Impact:**

Instead of a rigid assumption that  P(y|x)=1 under a dataset, is it possible to model the probability of y given x for the dataset with finetuned language models?

For the human experiments in line 261, I think the setting might be easy for humans since half of the candidates are randomly selected from other dialogues, and it would be easy for humans to get the appropriate next response given the context, even without the provided explanation.


Typos and grammar:
line 202: grammar
line 254: give -> gives




**Main Review:**

Originality: The method is a novel and unified extension of LIME and Shapley value tailored for the dialogue generation setting. The adaption for the dialogue setting is clear with mostly reasonable assumptions.

Quality: The experimental design is reasonable and complete. One drawback is that the experiments are done on only one social dialogue dataset while there are other types of dialogues such as task-oriented dialogues as well.

Clarity: The paper is written clearly and easy to follow. I learned a lot from it. One suggestion is that some important information is left in the appendix and if space permits in the final version, more details on the human experiment can be added to the main paper.

Significance: It's not clear if statistical tests were performed from the tables and figures. If the results are statistically significant, I think this work could be a good addition to the current local explanation framework.







**Time Spent Reviewing:**

1 hour

---

> ### Author Response · Authors · 2021-08-10
> **Authors' response**
>
> We thank the reviewer for their review. Please find our answers below:
> 1. **If results are statistically significant:** We randomly initialized the explanation model and perturbation three times. We did a z-test and a t-test with the null hypothesis that $LERG_L$ will perform worse than $LIME$ (and $LERG_S$ v.s. $Shapley$). The p-values turn out to be less than 0.001, indicating that the proposed method highly significantly outperforms the baselines. We will add the statistical significance test to our revision.
> 2. **Experiments on other types of dialogues:** We have run experiments on another dialogue generation benchmark, personachat (Zhang et al. 2018). This dataset is less socialized and more about personal things description. The results turn out to have a similar trend as DailyDialog. The $LERG_S$ consistently improves $Shapley$ in the automatic metrics. The $PPLC_R$ increases as the removal rate is higher. The $PPL_A$ achieves the best score when the additive rate is about 0.6, meaning that it remains the important and non-noisy input for prediction. We plan to include this experiment in the final manuscript. We also plan to experiment on task-oriented dialogue datasets.
> 3. **Some important information is left in the appendix and if space permits in the final version, more details on the human experiment can be added to the main paper:** Thank you. We will add more details of the experiments to the main paper.
> 4. **Instead of a rigid assumption that P(y|x)=1 under a dataset, is it possible to model the probability of y given x for the dataset with fine-tuned language models?** We currently do not adopt this approach for two reasons. First, if using a fine-tune language model for each dataset, we will need much more training time before we can explain the generation models. This means the inference time can be increased substantially. Second, a language model is another blackbox model, so it may induce more uncontrollable factors into the explanation method. However, this is an interesting idea and a potential future direction.
> 5. **For the human experiments in line 261, I think the setting might be easy for humans since half of the candidates are randomly selected from other dialogues, and it would be easy for humans to get the appropriate next response given the context, even without the provided explanation.**  Our response: To remove the bias of humans guessing the answers, we adopt a “random” baseline to simulate the cases when humans should get the response given the context. It turns out that most explanation methods can get much higher accuracy than the random one. We suppose that comparing with the random baseline can remove the worry.
> 6. **Typos and grammar: line 202: grammar line 254: give -> gives:**   Thank you. We will proofread the final manuscript carefully.

---

> > ### Comment · Reviewer_CScg · 2021-08-24
> > **Thanks for the response**
> >
> > My questions are mostly addressed. Thanks!

---

### Official Review · Reviewer_TNnK · 2021-07-15

**Rating:** 7
**Confidence:** 3

**Summary:**

The authors propose a method for “explaining” dialogue responses from sequence generation models; that is, determining what parts of the input influence or “explain” certain elements of the output. The method, LERG, is “learned” in one of two ways to assign a score for each input/output token pair indicating the measure of relevance, and ultimately to link the intent between the two. The authors propose that their method must satisfy three properties: unbiased approximation (LERG should explain benefits of picking response given the context); consistency (LERG should explain each generation step); and cause identification (LERG should sort input features by importance). The authors measure the effectiveness of the proposed method in both automatic evaluations and user studies. The automatic evaluations measure the **necessity** and **sufficiency** of the explained segments of the input text by capturing the model perplexity difference when stripping the segments and only including the segments, respectively, essentially measuring how accurate the method’s explanations are at capturing the relevant segments of the input context. Human evaluations display the segments to humans and ask if they can predict the gold response. Both evaluations show their explanation method significantly outperform both random baselines and other methods from the literature.

**Limitations And Societal Impact:**

See comments above

**Main Review:**

*Originality*: The authors claim to be the first to apply local explanation methods to dialogue response generation. It is indeed clear how the work differs from prior contributions; specifically, the authors note how their method can be seen as a combination of prior work in the area of controllable generation and explanation of sentiment classifiers.

*Quality*: The submission is very technically sound, with the equations derived in the text itself. The claims are very well supported, and the experimental results clearly outline the benefit of using the authors proposed method. I especially thought the method of human evaluation was a very appropriate way of measuring the effectiveness of the explanation method.

*Clarity*: The authors structure the paper appropriately, and presentation of results is done well and highlights the effectiveness of the method. However, The middle of the paper describing the methods involve several mathematical equations that are slightly cumbersome (I think some more details could be left to the appendix).

*Significance*: The authors propose a unique and quantifiable method for explanation of response generation models; this is very significant in the space of open-domain dialogue in any capacity where one would like to know why a model said what it said. Implications range from safety applications to control applications.

**Time Spent Reviewing:**

1.5 hours

---

> ### Author Response · Authors · 2021-08-10
> **Authors' response**
>
> We thank the reviewer for their review.  We are happy to answer any additional questions at any time.

---

### Official Review · Reviewer_Dmqp · 2021-07-16

**Rating:** 6
**Confidence:** 4

**Summary:**

This paper proposes a model-agnostic explanations model, local explanation of response generation (LERG), for dialogue response generation task. Due to the sequence-to-sequence natrual, previous works, which normal produces a single label as output, are no longer suitable for this task. This paper regards the
explanations as the mutual interaction of segments in input and output sentences. Specifically, it views the sequence prediction as uncertainty estimation of a human response and creates explanations by perturbing the input and calculating the certainty change over the human response. The experiment shows that the proposed approach extract both explicit and implicit relations between input and output segments.

**Main Review:**

Originality:
model-agnostic explanation models for sequence-to-sequence task are not a new task. [1] proposes methods to explain the black-box sequence-to-sequence model, including machine translaton and dialogue task. This paper first review popular model-agnostic explanation models, LIME and Shapley value, then reformulate these two methods into a novel and unified framework. In order to model input and output segment interaction, this paper proposes to use a two-dimensional matrix to measure the influence of of input and output. This is different from the above two methods. However, I am concerned that this is not a novel approach because [1] proposes a similar idea to use a bipartite graph whose nodes are tokens in the input and output seuqences and input tokens are only connected with the output tokens for explaining machine translation and dialogue. Furthermore, the algorithm proposed here only consider human annotated output sequence. Most of the equation derivations in this paper (i.e., Eq 8 - 10) are based on this strict assumption. Although this brings some theorical benefits (shown in Section 3.3, somewhat similar to the analysis in [2]), explaining human annotated output could be redundant because we can directly ask the annotators for the explaination. Finally, this paper follows optimization in LIME and Shapley value method to solve the proposed two-dimensional explainable models. This is the main novelty that I am aware of after reading this submission. To improve this submission, I believe that expanding the method to model generated response will be very helpful. In addition, although the authors claim that dialogue task is "challenging with its open-ended sentences and multiple acceptable responses", the whole algorithm design does not consider any spefical properties for dialogue task. It is also very interesting to see if this algorithm can be applied to other sequence-to-sequence tasks, such as machine translation and text summarization.

Quality & Clarity:
This paper presents a fairly completed work with various experiments, including automatic metrics and human evaluation. However, the authors do not provide weaknesses (e.g., what happens if we apply it to model generated response) of their proposed model. The paper includes significant amount of equation derivations in the main text, and some of them could be moved to supplementary materials (i.e., Eq 8). The author provides source code with this submission which should be helpful for reproducibility.

Significance:
Model-agnostic explanations model for seqence-to-sequence tasks should be interesting to most of people in text generation community. However, the algorithms proposed in this paper are similar to [1]. They both use two-demensional matrix to represent the interaction between input and output sequence. In addition, in Sec 5, the author compare to [1] by saying "these tasks are less complex than explaining general text generation models". I cannot agree on this because [1] is proposed to explain black-box sequence-to-sequence models.

**Time Spent Reviewing:**

6

---

> ### Author Response · Authors · 2021-08-10
> **Authors' response**
>
> We thank the reviewer for their review.  Please find our answers below:
> 1. **Difference from reference [1]:** We appreciate reference [1], our work differs in the following ways
>     1. **About the methods:** Reference [1] does a great job on their proposal of a bipartite graph to filter out the most important scores among all pairs for easy visualization. Our method works orthogonally by focusing on computing theoretically based interacted scores of inputs and outputs for response generation tasks. Combining these two methods is an interesting area of research.
>     2. **About the dialogue generation experiment settings:** Reference [1] focuses mostly on machine translation and does not discuss dialogue generation deeply. In the dialogue generation experiment, they trained a generative model with the common issue in dialogue generation, generic responses. This issue will result in models having the highest probability to predict generic, not-so-good responses such as “I don’t know”. This issue has made many recent models use top-k sampling on the predicted probability distribution. To mitigate the effect of the sampling process, we propose to explain how the model works before sampling. We further suggest that explaining the probability of a trained model to predict a human annotated response is more reasonable. Our proposal can ensure that:
>         1. We are explaining a reasonable response.
>         2. We do not rely on samples from a chosen sampling method (e.g., top-k sampling).
>         3. This method can be used to compare different models, since they use the same reference.\
> If one wants, we can also easily use this method for predicted responses. In that case, we suggest dividing the evaluation process into two stages. First, use the annotated responses to evaluate which explanation methods and models are better. Next, use the best explanation method and the generated response to analyze if a model’s rationale is proper.
>     3. **About the focused tasks:** In machine translation, there are additional confounding variables, such as the linguistic properties for selected language pairs, language divergence, typology, morphology, etc. In our monolingual setting, we have a better controlled study to investigate explanations with less confounding variables. Also the explanations of machine translation could be easier to evaluate and might possibly be reproduced using an alignment model, since machine translation is a task where most parts in the input and output can find direct relations. This was the reason we claim that the task is less complex in Sec 5. We will pay more attention to tone.
> 2. **Other sequence-to-sequence tasks:** We have run experiments on an abstractive summarization benchmark, XSum(Narayan et al. 2018). We explore this task since abstractive summarization shares the similar property, less word-to-word mapping in the inputs and outputs, with dialogue generation. It turns out to have a similar trend as DailyDialog. $LERG_S$ consistently improves $Shapley$ in the automatic metrics. The $PPLC_R$ will increase as the removal rate is higher. $PPL_A$ will achieve the best score when the additive rate is about 0.5, meaning that it remains the important and non-noisy input for prediction. We plan to include this experiment in the appendix of the final manuscript.
> 3. **Provide the weaknesses of the proposed model (e.g., generated response):** We discuss the weaknesses of the methods in section 4.5. Regarding the generated responses, we provide our answer in Q1.
> 4. **Move some equations to supplementary materials (i.e., Eq 8):** We will proofread and move parts to the appendix.

---

> > ### Comment · Reviewer_Dmqp · 2021-08-27
> > **Thank you for your reply**
> >
> > After carefully reading your responses,  my concern about the originality has been addressed, I have improved my score.

---

### Official Review · Reviewer_BdaH · 2021-07-19

**Rating:** 6
**Confidence:** 2

**Summary:**

This work proposes a method to locally explain the dialog response generation process. This method is motivated by the intuition that, such method should answer the question "which parts of the response are influenced the most by parts of the prompt". So the importance scores of every input-output segment pairs are calculated. They also proposed two metrics to evaluate explanations: "necessity" and "sufficiency". The former is defined as the perplexity change after removing top "salient" input features, and the latter is defined based on the intuition a complete explanation should recover model's prediction without the original input.

**Limitations And Societal Impact:**

See above concern regarding ngram-level explanation.
This reviewer doesn't think this work has negative societal impact. Actually, if reasonable explanations can be provided for machine learning generative models, it should have positive societal impact for people to understand the models.

**Main Review:**

this work focuses on an important problem: the explanation of generative models. This work proposes a model-agnostic tool to provide explanations, and such tools should be very helpful to the process to debug, understand and trust the model. Their pairwise importance scores also seem a reasonable method.

However, this reviewer believe this work can be improved if the following concerns can be resolved: The examples shown in Figure 4 are some token-to-token relation, this together with the error analysis seems to indicate that ngram-level segment (not token-level) can be sometimes important and not well captured by the model. Could the authors comment on this? is it easy to show ngram-to-ngram relation? does it perform well?

**Time Spent Reviewing:**

24

---

> ### Author Response · Authors · 2021-08-10
> **Authors' response**
>
> We thank the reviewer for their review.  Please find our answer below:
> - **How to show ngram-to-ngram relation?** We did two ways of phrase-level experiments.
>   1. In the first approach, we used parsed phrases, instead of tokens, as the x_i and y_j in the equations and obtained explanations through Algorithm 1.
>   2. In the second approach, we used the token-level explanations and averaged the scores of tokens in the same phrase parsed by an off-the-shelf parser (we used pyStatParser, an open-sourced python package).
>
> In both cases, the performance was similar to token-level methods.  However, we suppose that the basic unit of dialogues is hard to define, and ngram is not necessarily the basic unit. Therefore we choose tokens which can be flexibly bottom-up to different levels of units in this paper. The two methods above are possible ways of looking into phrase or n-gram level. We will discuss our experiments on phrase- / ngram- level experiments in the revision.

---

### Official Review · Reviewer_R5ns · 2021-08-03

**Rating:** 6
**Confidence:** 4

**Summary:**

This paper studies the model-agnostic explanations of dialogue response generation and proposes a new called local explanation of response generation (LERG). LERG extracts the sorted importance scores of every input-output segment pair from a dialogue response generation model. It views the sequence prediction as uncertainty estimation of a human response and creates explanations by perturbing the input and calculating the certainty change over the human response. The authors show the proposed LERG adhere to three properties of  an ideal explanation of text generation: (1) unbiased approximation, (2) intra-response consistency and (3) causal cause identification. The experiments on a popular benchmark DailyDialog empirically verify the effectiveness of the proposed approach.

**Limitations And Societal Impact:**

Yes, the authors have addressed this.

**Main Review:**

**Strengths:**

The black box nature of current end2end neural dialogue models hinders us from understanding the underlying reasoning process of these models. This paper introduces a novel method to explore explanation for dialog response generation models for the first time. Overall, I like this idea.

They mathematically formulate the task and generalizes local explanation methods towards sequence generation and show that our method adheres to the desired properties for explaining conditional text generation.

They introduce the corresponding evaluation methods to assess the explanations of dialogue response generation and empirical study demonstrate the effectiveness of proposed LERG.


**Weaknesses:**

Experiment conducted on only one dataset is somewhat a little weak. Have you tried the other popular benchmarks in the dialogue generation task, such as Reddit Corpus, Persona-Chat (Liu et al., 2020) and Topical-Chat (Gopalakrishnan et al., 2019)?

Section 4.2 and 4.3 are lack of the insightful analysis for the reported results. I suggest the authors to share more insights to readers here, rather than just reporting the plain experimental results.

It seems there is no clarification about the code release in the paper. Will you make your code publicly available to the community for the reproduction later?

Minor comments:

missing reference on controllable dialogue generation in the related work:

[1] Zhang et al. Learning to control the specificity in neural response generation. ACL 2018.

[2] Xu et al. Neural response generation with meta-words. ACL 2019.

[3] See et al. What makes a good conversation? how controllable attributes affect human judgments. arXiv 2019.

[4] Smith et al. Controlling Style in Generated Dialogue. arXiv 2020.

[5] Wu et al. A controllable model of grounded response generation. AAAI 2020.

[6] Madotto et al. The Adapter-Bot: All-In-One Controllable Conversational Model. arXiv 2020.

[7] Yang et al. Progressive Open-Domain Response Generation with Multiple Controllable Attributes. IJCAI 2021.

[8] Gupta et al. Controlling Dialogue Generation with Semantic Exemplars. NAACL 2021.











**Time Spent Reviewing:**

6

---

> ### Author Response · Authors · 2021-08-10
> **Authors' response**
>
> We thank the reviewer for their review.  Please find our answers below:
> 1. **Other popular benchmarks in the dialogue generation task:** We have run experiments on another dialogue generation benchmark, personachat (Zhang et al. 2018). It turns out to have a similar trend as DailyDialog. The $LERG_S$ consistently improves over $Shapley$ in the automatic metrics. The $PPLC_R$ increases as the removal rate is higher. The $PPL_A$ achieves the best score when the additive rate is about 0.6, meaning that it remains the important and non-noisy input for prediction. We plan to include this experiment in the final manuscript.
> 2. **Share more insightful analysis for the reported results:** Yes, we plan on furthering our analysis on the reported results if additional pages are allowed for the camera ready. In particular, we will expand the analysis on all methods' performance; for example the attention based method performs similarly to random, signifying that it is not learning meaningful explanations. We had to cut the analysis down due to the page limit. We will also add additional analysis about the computational costs and the preliminary study of the correlation with dialogues engagement.
> 3. **About the code release in the paper:** We will open-source our implementation. This is currently mentioned in the appendix. We will move it to make it more visible in the main article with the link.
> 4. **Missing reference on controllable dialogue generation in the related work:** We will add these references to the related work and discussion section.

---

### Decision · Program_Chairs · 2021-09-27

**Decision:**

Accept (Poster)

**Comment:**

The paper studies the problem of generating model-agnostic explanations of dialog responses. The proposed method (LERG) estimates importance scores between every input-output segment pair by exploring perturbations of the input. LERG uses two optimization variations (Shapley value and LIME) originally proposed for classification and newly extended to sequence-to-sequence problems. The paper also comes with theoretical justifications, showing that LERG has the following desirable properties: unbiased approximation, consistency, cause identification (proofs are provided in the appendix). While the paper is evaluated only on dialogue, the methods of this paper seem applicable to other tasks such as machine translation.

The paper addresses a very important task, as the black-box nature of most current conversational AI systems is likely hindering our understanding of what goes wrong and what could be improved. Reviewers found the approach to be well motivated, technically sound, and the presentation of the work is quite clear. The only two concerns are:

* Alvarez-Melis and Jaakkola (2017) [1] presents a somewhat similar model-agnostic explanation generation method for sequence-to-sequence models. That said, there are some technical differences, and the paper offers technical contributions relative to [1]: the adaption of established model-agnostic explanations methods to seq2seq problems (LIME and Shapley value), and theoretical justifications (unbiased approximation, etc.). Furthermore, [1] is mainly focused on translation and its application to dialog seems a bit preliminary (the authors of [1] seem frank about that as they call their system “mediocre”). The few examples of dialog responses in [1] seem to suffer from a lack of diversity (“I don’t know”, etc.).

* The paper performs all experiments on gold responses, which is a bit unrealistic. The authors provide some good justifications (i.e., “we are explaining a reasonable response”, mitigating the effects of the sampling process typically used during inference, evaluating on the same gold outputs makes results more comparable), but I think evaluating both on gold and generated responses would have made the paper stronger.

Compared to previous work (e.g., [1]), the paper offers some empirical contributions: evaluation of "necessity" and "sufficiency" (“How is the model influenced after removing explanations?” and “How does the model perform when only the explanations are given?”, respectively). The paper also offers a user study to evaluate the effectiveness of LERG with real users, and the improvement over several baselines seems quite significant.